# Genome-Wide Identification of Mango (*Mangifera indica* L.) MADS-Box Genes Related to Fruit Ripening

**Bin Zheng** [1,2], **Songbiao Wang** [2], **Hongxia Wu** [2], **Xiaowei Ma** [2], **Wentian Xu** [2], **Kunliang Xie** [2], **Lingfei Shangguan** [1] **and Jinggui Fang** [1,*]

1. Horticulture Department, Nanjing Agricultural University, Nanjing 210095, China; zhengbin@catas.cn (B.Z.)
2. Key Laboratory of Tropical Fruit Biology, Ministry of Agriculture and Rural Affairs, South Subtropical Crops Research Institute, Chinese Academy of Tropical Agricultural Sciences, Zhanjiang 524091, China
* Correspondence: fanggg@njau.edu.cn

**Abstract:** MADS-box genes play a crucial role in fruit ripening, yet limited research has been conducted on mango. Based on the conserved domains of this gene family, 84 MADS-box genes were identified in the mango genome, including 22 type I and 62 type II MADS-box genes. Gene duplication analysis revealed that both tandem duplication and segmental replication significantly contributed to the expansion of MADS-box genes in the mango genome, with purifying selection playing a vital role in the segmental duplication events within the MiMADS gene family. Cis-acting element analysis demonstrated that most MiMADS genes were hormonally regulated and participated in the growth, development, and stress resistance of mango fruit. Moreover, through expression pattern analysis and phylogenetic tree construction, we identified six MiMADS genes belonging to the SEP1 subfamily and two belonging to the AG subfamily as potential candidates involved in mango ripening regulation. Notably, Mi08g17750 and Mi04g18430 from the SEP1 subfamily were identified as key regulators inhibiting mango fruit maturation; their interaction network was also analyzed. These findings provide a foundation for further investigation into the regulatory mechanisms underlying mango ripening.

**Keywords:** genome; MADS; gene family; fruit maturation



## 1. Introduction

MADS-box genes constitute one of the largest families of plant transcription factors (TFs) [1]. Structurally, nearly all MADS-domain proteins possess a highly conserved DNA binding domain called the MADS box at the N-terminus, consisting of approximately 60 amino acid residues. This domain recognizes and binds to CarG boxes (CC[A/T]6GG), thereby regulating downstream gene transcription [2]. From an evolutionary perspective, they can be categorized into two phylogenetically distinct groups: type I and type II [3]. Generally, type I MADS genes exhibit simple structures with one to two exons, and limited research has been conducted on their functions. In plants, the type-I MADS box TFs are further categorized into three groups based on the M domain of the encoded protein: Mα, Mβ, and Mγ. The type II genes are commonly referred to as MADS intervening keratin-like and C-terminal (MIKC) genes due to their four distinctive domain structures: the M domain, the K domain, the I (intervening) domain, and the variable C (C-terminal) region [4]. Based on differences in their domain structure, MIKC-type MADS-box genes have been divided into MIKC* and MIKC^C types [5]. MIKC* MADS-box genes feature an altered protein domain structure potentially resulting from a duplication of exons encoding the K domain subregion [6]. According to their phylogenetic relationships, MIKC^C-type MADS-box genes can be further subdivided into distinctive subfamilies, such as AG, AGL6, AGL12, AGL15, ANR1, AP1, AP3, FLF, P1, TT16, SEP1, SOC1, and SVP, based on their phylogeny.

MADS-box proteins are crucial transcription factors involved in the regulation of signal transduction [7], stress responses [8,9], and various aspects of plant development, including

vernalization and flowering time control [10,11], floral organ (carpels, stamens, sepals, and petals) development [12–16], carpel identity and pollen development [17,18], reproductive development [19], seed development and pigmentation [20–23], root development [24,25], formation and dehiscence of fruit [26–28], and endosperm development [29]. In recent years, numerous MADS box TFs have been discovered to be associated with fruit ripening [30]. Tomato SiRIN has emerged as a key factor in delaying fruit ripening, and it has been extensively investigated [31]. It was reported that SiRIN protein influences gene expression by binding to CArG box sequences present in the promoter regions of several ripening-related genes [32,33]. Furthermore, SiRIN interacted with other MADS-box proteins (FUL1, FUL2, TAGL1, TAG1, MBP21, and TDR5) to regulate fruit ripening [34]. Subsequently, some other tomato MADS box TFs, such as TAGL1 [35], SiMADS1 [36], SiFYFL [37], FUL [38], FUL1, FUL2 [39], SiMBP8 [40], and SiCMB1 [41], were also confirmed to be able to regulate tomato fruit ripening through fine-tuning ethylene biosynthesis and the expression of ripening-related genes. In addition, MADS-box proteins implicated in the regulation of fruit ripening have also been identified in strawberry [42,43], banana [44–46], papaya [47], citrus [48], and other fruits.

Mango (*Mangifera indica* L.) is a fruit of significant economic importance in the world, but it has a limited shelf life after harvest. However, the length of the development period varies greatly among different varieties. To prolong the availability of fresh mangoes, extremely early maturity and extremely late maturity have become key indicators for breeding new mango varieties. Previous studies have reported various biochemical changes during mango fruit ripening, including increased respiration, ethylene production, fruit softening, chlorophyll degradation, carotenoid synthesis, and other metabolic activities leading to changes in carbohydrates, organic acids, lipids, phenolics, and volatile compounds [49]. Nevertheless, the regulation mechanism underlying early maturing and later maturing in different mango varieties remains unexplored. MADS-box TFs are known to play a role in regulating fruit ripening. In this study, based on the mango genome data, the MADS box TFs of mango were identified and analyzed, and the MADS box TFs related to fruit ripening were screened. These findings lay a foundation for further elucidating the molecular mechanisms governing early and late maturity.

## 2. Materials and Methods

### 2.1. Identification and Features Analysis of MADS-Box Family in Mango

We obtained mango genome sequences, proteome sequences, and a GFF file from Wang [50] and retrieved the Arabidopsis MADS-box genes from The Arabidopsis Information Resource (TAIR) (http://www.arabidopsis.org/ (accessed on 22 April 2022)) databases. Two strategies were employed to identify the MADS-box transcription factor family. Firstly, the hidden Markov model (HMM) profile of SRF-TF domains (PF00319) and K-box domains (PF01486) from the Pfam database (Pfam 32.0, http://pfam.xfam.org/ (accessed on 22 April 2022)) were used as queries to identify MADS-box sequences with HMMER version 3 [51] against the mango genome with a threshold of e-value $\leq 1 \times 10^{-5}$. Additionally, 109 previously identified MADS protein sequences of Arabidopsis were used as queries to search against the mango genome using the BLAST program of TBtools [52] with an e-value of $1 \times 10^{-5}$. Subsequently, all potential candidate proteins were submitted to the Pfam database and the National Center for Biotechnology Information (NCBI) Conserved Domain Search (https://www.ncbi.nlm.nih.gov/Structure/cdd/wrpsb.cgi (accessed on 1 May 2022)) and Tbtools Batch SMART for confirmation of the presence and completeness of the MADS domain. Candidate genes lacking the MADS domain were re-annotated using the BLAST tool of TAIR. The physical and chemical properties of mango MADS-box (MiMADS) genes were predicted using the online tool ProtParam from ExPASy (https://web.expasy.org/protparam/ (accessed on 1 May 2022)). Subcellular localization prediction results for MiMADS genes were obtained using the tools from the WoLF PSORT website [53].

## 2.2. Phylogenetic Tree, Gene Structure, and Conserved Motif Analysis of MiMADS Genes

In order to gain a comprehensive understanding of the phylogenetic relationships, and further classify type I and type II MADS-box genes, the phylogenetic tree of mango and Arabidopsis MADS proteins was constructed using the maximum likelihood method of the MEGA7 software package according to the similarity of the full-length amino acid sequence. Based on the relationships between MiMADS and AtMADS proteins and the classification scheme of AtMADS, all of the identified MiMADS genes were categorized into distinct groups. The gene structures of MiMADS genes were obtained by comparing the open reading frames (ORFs) with the genomic sequences, which were displayed using the gene structure display server (GSDS). Additionally, conserved motifs of MiMADS proteins were investigated using the MEME program (http://meme.nbcr.net/meme/cgi-bin/meme.cgi (accessed on 5 May 2022)), with the parameters set as follows: site distribution (zero or one occurrence per sequence), motif number (20), and motif width (between 6 and 200 wide). Finally, obtained motifs were annotated using the NCBI Conserved Domain Search service (CD-Search) program [54].

## 2.3. Chromosomal Distribution, Gene Duplication, and Ka/Ks Analysis of MiMADS Genes

The chromosomal locations of MiMADS proteins were determined according to Wang [50], and the TBtools [52] software package was utilized for constructing linkage maps of MiMADS proteins. Gene duplication events of MADS-box genes in mango were also investigated using Tbtools. Syntenic analysis of MADS genes in mango, *Arabidopsis thaliana*, and *Citrus sinensis* was performed using the TBtools software, which embedded MCscan X software under default parameters [52]. The genomes of *Arabidopsis thaliana* and *Citrus sinensis* were downloaded from the Arabidopsis (https://www.arabidopsis.org/ (accessed on 10 May 2022)) and citrus (http://citrus.hzau.edu.cn/orange/ (accessed on 10 May 2022)) databases, respectively. Furthermore, the Nei–Gojobori method within TBtools was employed to calculate the non-synonymous substitution rates (Ka) and synonymous substitution rates (Ks) of MiMADS proteins.

## 2.4. Cis-Element Analysis of MiMADS Proteins

The 2000 bp sequences upstream of the start codon of each MiMADS gene were extracted to predict cis-acting elements using the PlantCARE database (http://bioinformatics.psb.ugent.be/webtools/plantcare/html/ (accessed on 6 May 2022)) [55]. In this study, we specifically focused on selecting cis-elements associated with hormonal and environmental responses.

## 2.5. Interaction Networks Analysis of MiMADS Proteins

The protein–protein interaction networks of MADS proteins in *Arabidopsis thaliana* and tomato were selected as references, respectively, and the interaction relationship between MiMADS proteins was further analyzed through homologous comparison. The online website STRING 11.0 (https://string-db.org/ (accessed on 1 November 2022)) was used to display the protein–protein interaction network.

## 2.6. Plant Materials Collection and Expression Analyses Based on Publicly Available RNA-seq

The samples utilized in this experiment conform to Li [56]. Two mango varieties (namely, the early maturing Tainong-1 and the late maturing Renong-1) were employed as plant materials. All plants used in this study were cultivated at the South Subtropical Crop Research Institute in Zhanjiang, China (21°10′2″ N; 110°16′34″ E). Samples of Tainong-1 from four different stages were collected on March 27th (young fruit stage), April 27th (fruit enlargement stage), June 4th (green mature stage), and June 14th (full-ripe stage, one week after harvesting green, mature fruits). Meanwhile, samples of late-maturing variety Renong-1 were collected on March 27th (young fruit stage), May 28th (fruit enlargement stage), July 6th (green mature stage), and July 13th (full-ripe stage, one week after harvesting green mature fruits). The Illumina transcriptome raw data of two varieties at four different stages,

each with three biological replicates, were downloaded from NCBI BioProject PRJNA 629065. Clean reads were retrieved after removing low-quality (containing >50% bases with a Phred quality score < 20) reads and those with unknown nucleotides (more than 1% ambiguous residues N) using the FastQC tool (http://www.bioinformatics.babraham. ac.uk/projects/fastqc/ (accessed on 1 April 2022)). The clean reads were aligned to the reference genome [50] using Hisat2 (v2.1.0) software with default parameters [57]. Gene expression levels in each sample were estimated using fragments per kilobase of transcript per million fragments mapped (FPKM) values using Cuffdiff (v2.1.1). Multiple hypothesis testing correction of the hypothesis test probability (*p* value) using the Benjamini–Hochberg method was also needed to obtain the false discovery rate (FDR). Genes meeting the criteria of *p*-value < 0.05 and an absolute log2 (Fold Change) $\geq$ 1 were identified as differentially expressed genes.

## 3. Results

### 3.1. Identification of MADS Proteins in Mango

The identification of all potential MADS members in the mango genome was accomplished using two approaches: BlastP search and Hidden Markov Model (HMM) search. As a result, a total of 84 MiMADS proteins were successfully identified. Subsequent analysis confirmed that all identified MiMADS proteins contained conserved domains, such as the MADS domain, SRF-TF domain, or K-box domain (Figure 1). Out of the total MiMADS proteins, 24 MiMADS proteins had only MADS or SRF-TF domains, 3 had only K-box domains, and 57 encoded for both domains (Figure 1). The length of these 84 mango MiMADS proteins ranged from 61 (Mi15g00500.1) to 395 (Mi01g22630.1) amino acid residues, with relative molecular masses ranging from 6.96 (Mi15g00500.1) to 44.21 (Mi01g22630.1) kDa, while their isoelectric points fell within the range of 4.96–10.68 (Table S1). Out of the 84 MiMADS proteins, only 8 with an instability index below 40 were stable proteins, while the rest were unstable proteins with indices above 40. The aliphatic index (AI) was used to assess protein thermostability and ranged from 74.94 to 102.21 for MiMADS proteins. Except for Mi10g15870.1, all other MiMADS had a grand average of hydropathicity (GRAVY) less than zero and were hydrophilic proteins. Predicted subcellular localization results indicated sixty-six MiMADS proteins localized in the nuclear region, with eight in chloroplasts, seven in cytoplasm, and three in mitochondria.

### 3.2. Phylogenetic Relationships of the MiMADS Proteins

To investigate the phylogenetic relationships between MADS family proteins of mango and those in Arabidopsis, an unrooted maximum likelihood tree was constructed using full-length MADS-box proteins from both species with the assistance of MEGA6 software (Figure 2). According to the classification scheme of Arabidopsis MADS-box proteins based on previous research [58], all identified MiMADS genes were categorized into two groups: type I (22) and type II (62). Among the 22 type I genes, 16 were classified as members of the Mα subgroup, while 4 belonged to the Mβ subgroup, and 2 fell into the category of the Mγ subgroup. As for type II MiMADS genes, a majority of them (53) were assigned to subgroup MIKC$^C$, whereas 9 remaining members were classified into subgroup MIKC*. Based on Athaliana's classification system for its own set of MADS-box genes [4], mango MIKC$^C$ proteins can be further subdivided into thirteen subfamilies, along with their corresponding orthologous MADS proteins from Arabidopsis. Among these subfamilies, P1 and FLF each contained only one MADS protein; however, SOC1 had eleven MiMADS members, followed by seven for SEP1, six for AP1, five each for SVP, AG, and ANR1, three each for AGL15 and AGL6, and two each for TT16, AGL12, and AP3.

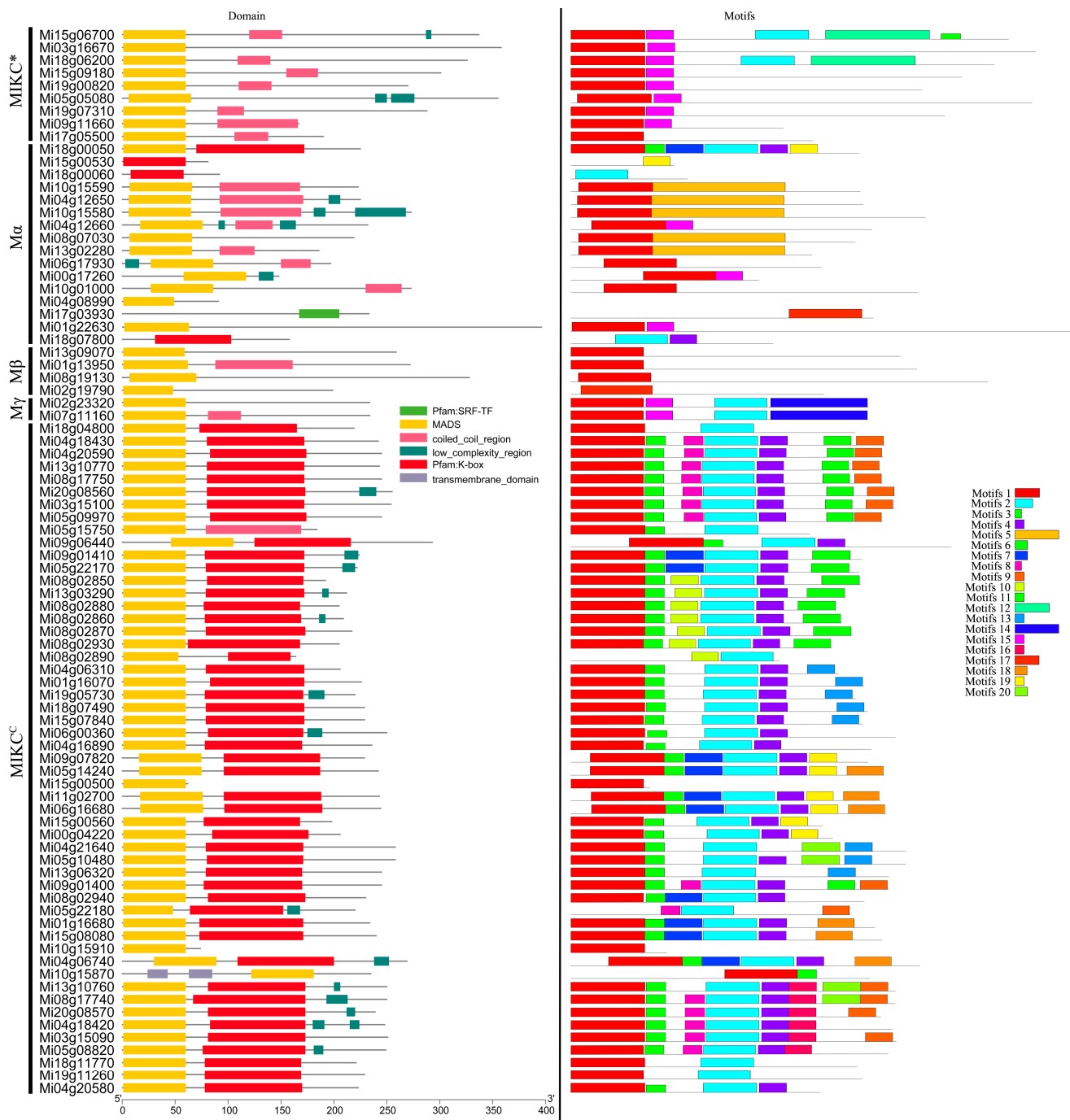

**Figure 1.** Domain and motif analyses of MADS proteins in mango.

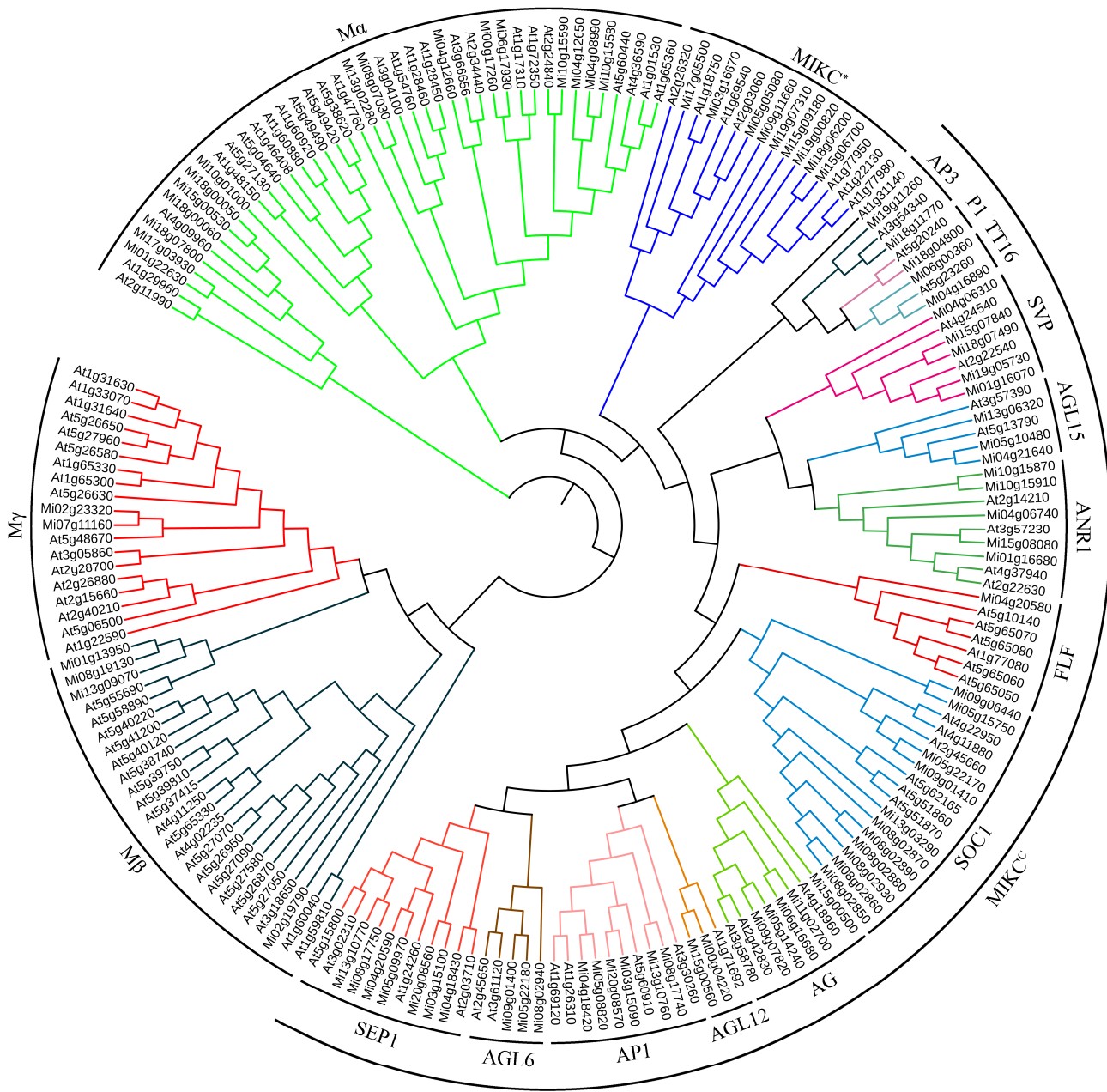

**Figure 2.** Phylogenetic tree analysis of mango and *Arabidopsis thaliana* MADS proteins.

### 3.3. Gene Structure and Conserved Motifs in MiMADS Proteins

The exon–intron organizations of the 84 MiMADS genes were also examined in an attempt to gain a better understanding of their structural evolution. As depicted in Figure S1, only four out of the eighty-four MiMADS genes consisted of a single exon, and all of the intronless genes were clustered within the type I group, including two Mα subgroup genes and all of the Mγ MADS subgroup genes. On the other hand, the remaining MiMADS genes exhibit varying numbers of exons ranging from two to eight. It is noteworthy that most of the genes containing seven to eight exons fall into the MIKC$^C$ classification. This observation suggests that similar exon–intron organizations exist among MiMADS genes within the same group and highlights that gene structure may hold significance for gene evolution and function. Based on previous research conducted on rice, it has been concluded that intron loss occurs at a faster rate than intron gain following segmental duplication [59]. Therefore, it can be inferred that most II-type MADS-box genes likely retain their original

structures, while many I-type group members appear to have predominantly undergone duplication through segmental duplications followed by subsequent intron loss.

The motif distribution of MiMADS proteins was analyzed using the MEME online software tool. A total of 20 conserved motifs, designated as motifs 1–20, were identified in the set of 84 MiMADS proteins (Figure 1). The detected motifs were annotated using NCBI CD search. As anticipated, motif 1 corresponded to the typical MADS-domain and was found in the majority of MiMADS proteins. Motif 2, which specifies the K domain, was prevalent among MIKC$^C$ type proteins, which were the second most conserved domain and essential for protein–protein interactions among MADS-box transcriptional factors [60]. Additionally, it was observed that most MiMADS proteins within a particular group shared a similar motif composition. For instance, motifs 5 were exclusively present in group M$\alpha$; motif 14 was solely found in groups M$\gamma$; motifs 12 were only detected in group MIKC*; whereas motifs 6, 8, 9, 11, 13, 16, 18, and 20 occurred exclusively within group MIKC$^C$. This suggests functional similarity within each respective group.

Previous studies have demonstrated that the presence of a K-box domain is exclusive to type II MADS-box genes [61]. However, our study reveals that four type I genes (Mi18g00050.1, Mi15g00530.1, Mi18g00060.1, and Mi18g07800.1) also possess a K-box domain. Additionally, simpler functional roles are associated with lower domain complexities. For instance, limited information exists regarding the impact of MIKC*, M$\gamma$, M$\alpha$, and M$\beta$ on plant growth and development due to their relatively uncomplicated structures consisting of one to six motifs. Conversely, the majority (44 out of 53) of the MIKC$^C$ group exhibit intricate structures with 4 motifs and play diverse roles in plant growth and development.

### 3.4. Chromosomal Localization and Gene Duplication of MiMADS Genes

To investigate the genomic distribution of MADS-box genes and explore their evolutionary patterns in the context of whole genome duplication, we employed mapping coordinates to assign each gene onto the mango genome (Figure 3). Our findings revealed that 82 MiMADS proteins were unevenly localized on 17 chromosomes based on a previously published report of mango [50]. Meanwhile, two MADS-box genes were traced on the unanchored scaffolds, which could be attributed to the incomplete assembly of the mango genome. The highest number of MADS-box genes (11) was located on chromosomes 4 and 8. They were followed by chromosome 5, which had eight MiMADS genes. Additionally, both chromosomes 15 and 18 harbored seven genes, while six were identified on chromosome 13. Chromosomes 9 and 10 each had five MADS-box genes. The numbers of MiMADS genes located on chromosomes 1, 2, 3, 6, 7, 11, 17, 19, and 20 were less than five. Meanwhile, chromosomes 12, 14, and 16 had no MADS-box genes.

Previous studies have demonstrated that gene duplications, including both tandem and segmental duplications, play a pivotal role in the expansion of the MADS-box gene family and significantly contribute to the proliferation of MADS-box genes across various plant species [62,63]. In mango, we identified 34 MADS-box genes (40.48%) located within 14 clusters as tandem duplicated genes consisting of 7 M$\alpha$ and 27 MIKC$^C$ group MiMADS genes. Among these clusters, three were located on chromosome 4, two were located on chromosomes 8 and 10, respectively, while one cluster was found on each of chromosomes 3, 5, 9, 13, 15, 18, and 20. The largest cluster comprised seven MADS-box genes located on mango chromosome 8. Interestingly, while the majority of clusters consisted of genes of the same types, we detected a cluster on chromosome 15 that contained different types of genes. Additionally, we identified 47 pairs of MiMADS genes located on segmental duplicated genome blocks, including six MIKC*, three M$\alpha$, two M$\gamma$, and thirty MIKC$^C$ group MiMADS genes. To reveal the direction of evolution, we estimated the Ka/Ks ratios (where Ka represents nonsynonymous substitutions per site and Ks represents synonymous substitutions per site) for these duplicated gene pairs. The Ks values for 7 MADS gene pairs were NaN; however, the Ka/Ks ratios for the remaining 40 MADS gene pairs were less than 1, ranging from 0.08 to 0.64 with an average value of 0.21 (Table S2). These findings

indicate that purifying selection played a crucial role in the segmental duplication events within the MiMADS gene family.

To establish the orthologous relationships of MiMADs, we conducted a comparative analysis of the genomic physical locations of MADS genes in *Mangifera indica*, *Citrus sinensis*, and *Arabidopsis thaliana* genomes. Based on the collinearity diagram (Figure S2), we identified 60 orthologous gene pairs between mango and Arabidopsis as well as 89 orthologous gene pairs between mango and sweet orange, indicating significant genetic similarities.

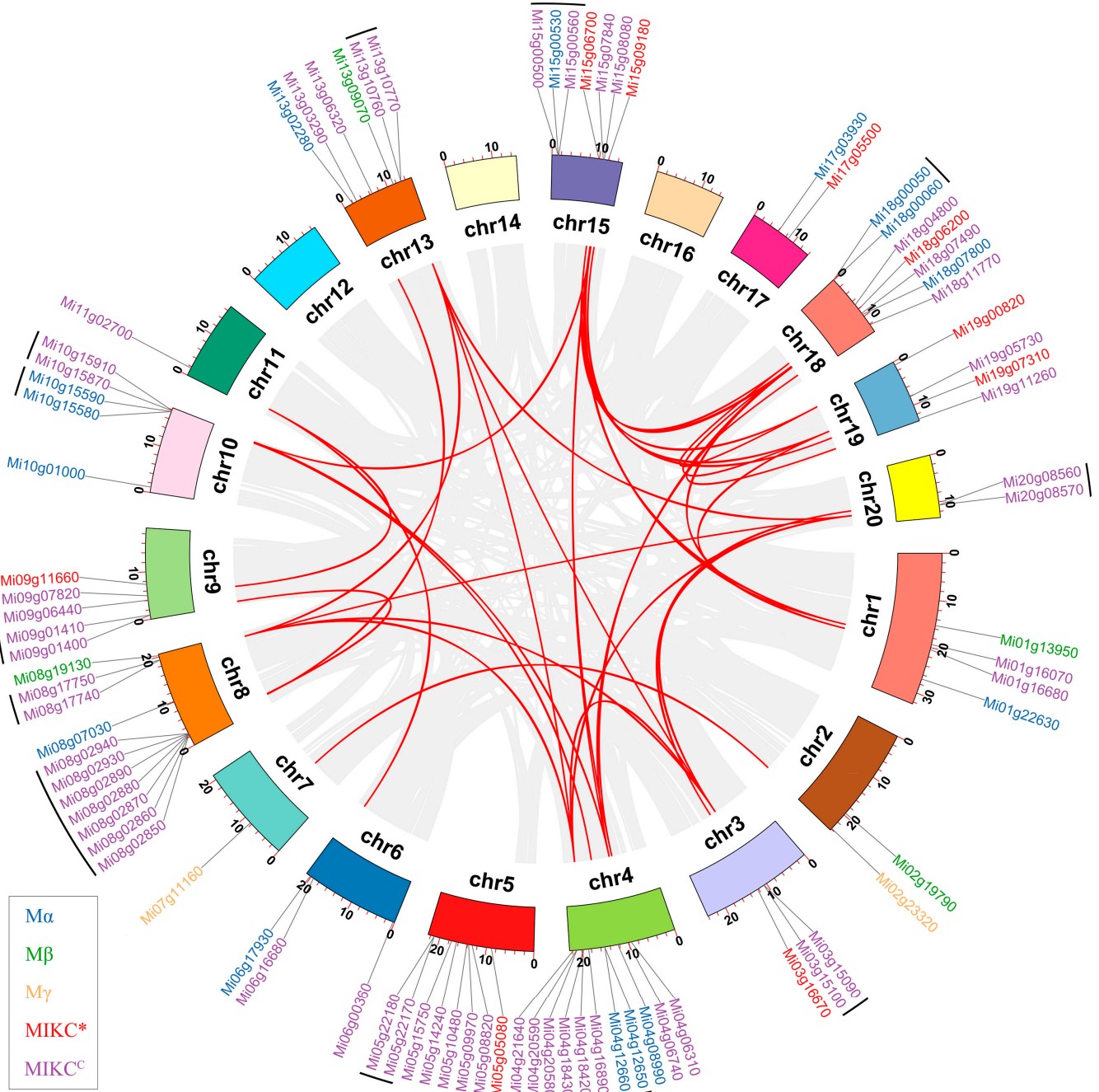

**Figure 3.** Chromosomal localization and gene duplication of MiMADS proteins. The lines inside the circos plot indicate segmental duplications; the short black lines outside the circos plot indicate tandem duplications.

### 3.5. Promoter Analysis of MADS Gene Family

Most of the well-characterized MADS-box genes are involved in plant growth and responses to hormones or environmental stimuli, such as photoperiod and temperature. Analyzing the promoter region features of MiMADS proteins will enhance our understanding of the expression patterns of MiMADS genes.

The cis-acting elements within the 2000 bp region upstream of the MADS gene family were investigated using the PlantCARE online tool [55]. Various types of elements were identified: hormone responsive, environmental stress related, promoter related, site binding related, developmental-related elements, and others (Figure 4, Table S3). Among these elements, we statistically analyzed the hormone-responsive elements present in the MADS gene family of mango and identified as-1 and TCA related to salicylic acid responsiveness in 49 MiMADS proteins. ERE and TGACG-motif related to ethylene and MeJA responsiveness, respectively, were identified in more than 40 MiMADS proteins. ABRE and TGA-element associated with the abscisic acid responsiveness and auxin-responsive element were found in 27 and 13 MiMADS proteins, respectively. Additionally, we attracted attention to the cis-elements responding to environmental stressors and identified stress-responsive (WUN-motif, TC-rich repeats, ARE, AAGAA-motif, LTR, and STRE) cis-elements in 77 MiMADS family members, and light-responsive boxes (GT1-motifi, ACE, and G-box) in 54 MiMADS proteins. We also found 12 and 10 MYB binding site MBS and MBR, respectively. In summary, these results suggest that diverse cis-acting elements related to hormones and stress response regulate the functional expression of most MADS genes in mango fruit.

### 3.6. Screening of Early- and Late-Maturation-Related MADS and Expression Patterns Analysis

The RNA-seq data of two mango varieties were remapped to the mango genome. It was observed that out of the 84 MiMADS proteins, 79 genes were expressed in fruit, and 55 genes exhibited differential expression across various stages of fruit development and ripening. Among these 55 differentially expressed MiMADS genes, the average FPKM value of 34 MiMADS genes in two varieties at different stages exceeded 1. Heat map analysis (FPKM value, log10 scale) was further conducted on these 34 MADS genes. As depicted in Figure 5, distinct variations were observed in the expression patterns of different genes during various developmental stages between the two varieties. Notably, nine MiMADS genes displayed an average FPKM value greater than 100 across different stages in both varieties. Among the nine high-expression differential MADS genes, five MiMADS genes with highest mean FPKM values (Mi20g08560, Mi03g15100, Mi05g09970, Mi13g10770, and Mi04g18430) and Mi08g17750 belonged to the SEP1 subfamily. Additionally, two genes (Mi05g14240 and Mi06g16680) belonged to the AG subfamily. One gene (Mi20g08570) belonged to the AP1 subfamily.

Phylogenetic tree analysis was conducted on these 34 differentially expressed MADS genes and other previously reported MADS genes associated with maturity (Figure S3). The results revealed that Mi13g10770, Mi08g17750, Mi05g09970, Mi20g08560, Mi03g15100, and Mi04g18430 were grouped together with apple's MdMADS8 and MdMADS9, banana's MaMADS1 and MaMADS2, as well as tomato's Rin gene. Additionally, Mi18g00050, Mi18g00060, Mi06g16680, Mi09g07820, and Mi05g14240 were clustered alongside MaMADS7, TAG1, and TAGL1.

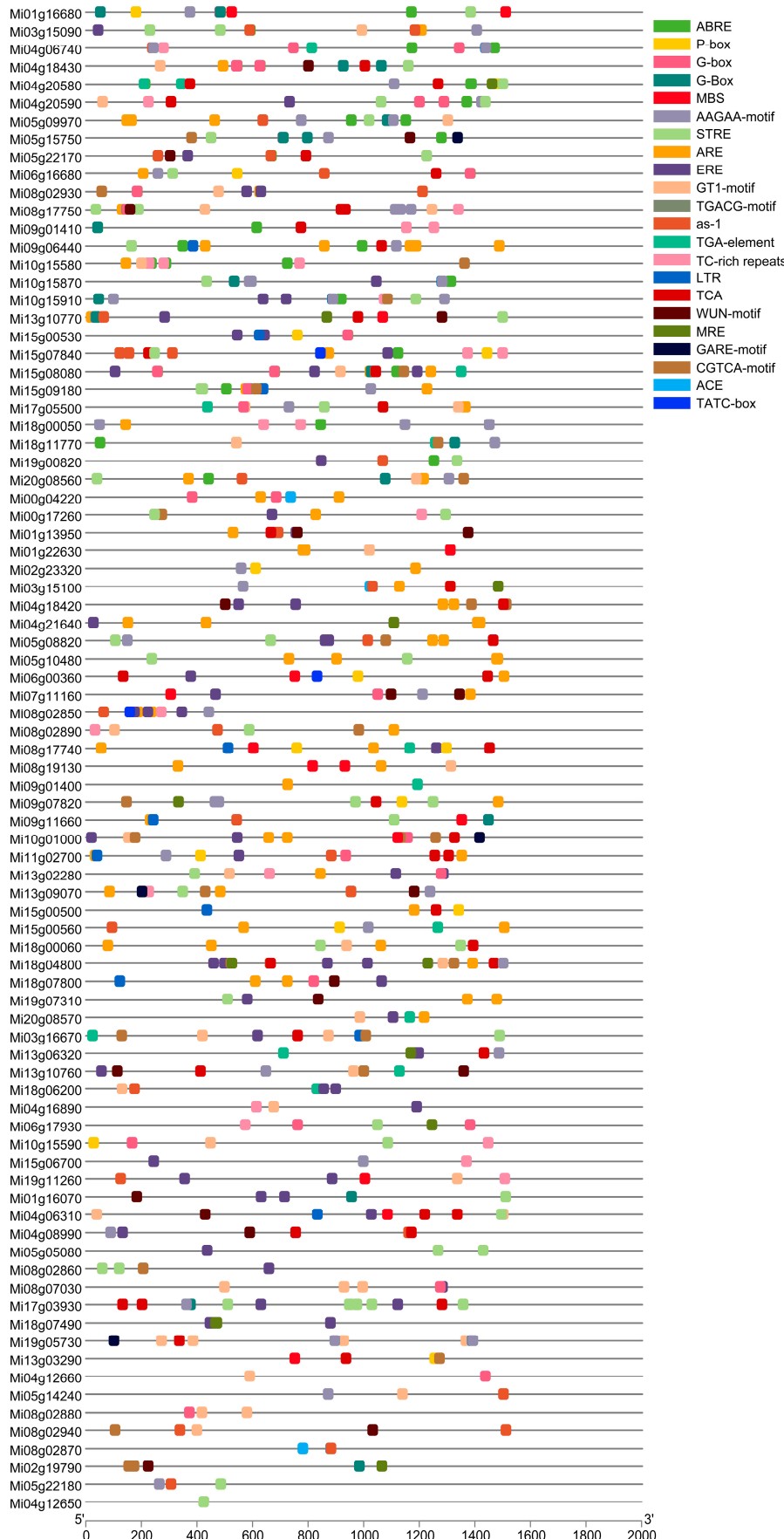

**Figure 4.** Cis-acting elements analysis of MiMADS genes.

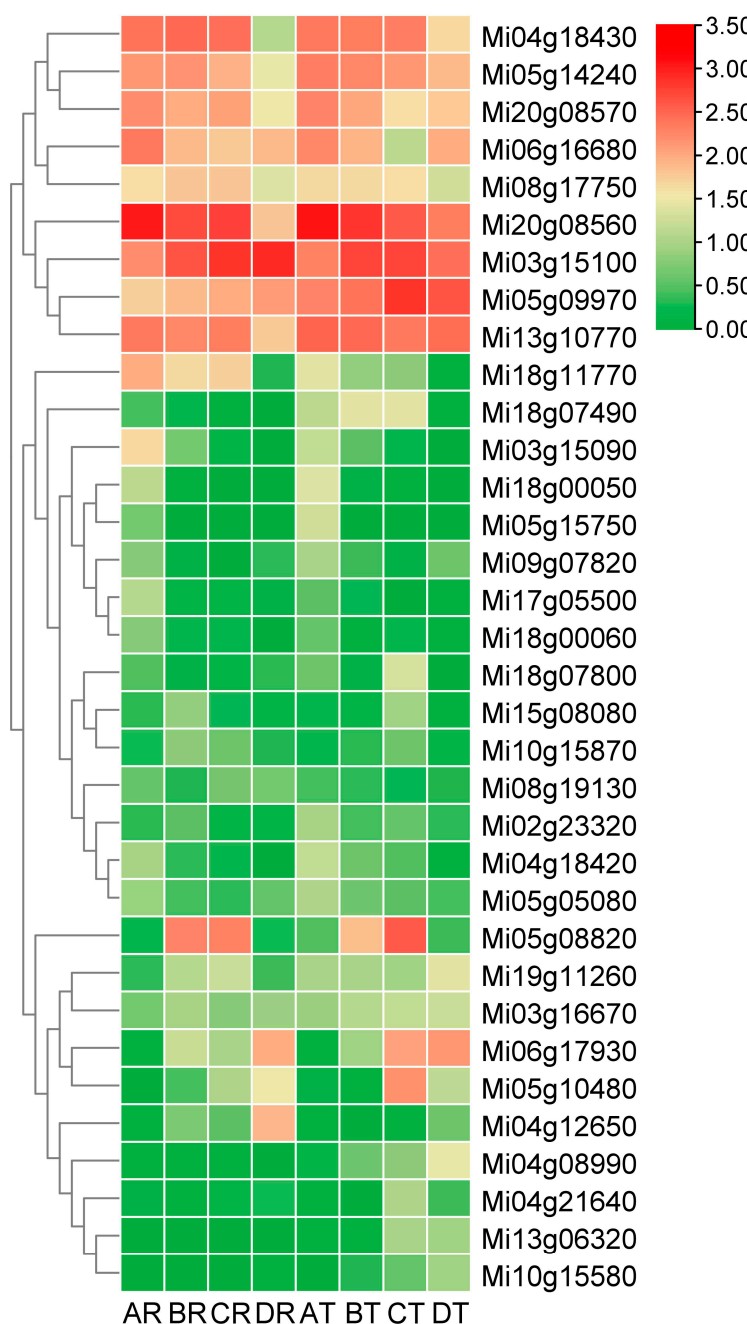

**Figure 5.** Heatmap of MiMADS FPKM value scaled by log10. AR, BR, CR, and DR and AT, BT, CT, and DT represent four different stages (young fruit stage, fruit enlargement stage, green mature stage, and full-ripe stage) of varieties Renong-1 and Tainong-1, respectively.

The comprehensive analysis of the phylogenetic tree and the average FPKM value (average FPKM value > 10) suggested that Mi20g08560, Mi03g15100, Mi05g09970, Mi13g10770, Mi04g18430, and Mi08g17750 from the SEP1 subfamily, as well as Mi05g14240 and Mi06g16680 from the AG subfamily, were key MiMADS genes associated with mango ripening. These results were closely aligned with the report that the AG and SEP subfamilies were the key regulators of fruit development and ripening [64]. Among them, two genes (Mi03g15100 and Mi13g10770) exhibited contrasting expression patterns during the postharvest period between Renong-1 and Tainong-1 varieties (Figure 6). Specifically, the expression levels of Mi03g15100 increased while those of Mi13g10770 decreased during the postharvest period of the Renong-1 variety. In contrast, they showed a decreasing trend for Mi03g15100

but an increasing trend for Mi13g10770 during the postharvest period of the Tainong-1 variety. The expression patterns of Mi05g09970 and Mi06g16680 in the variety Tainong-1 also exhibited contrasting trends, with an initial increase followed by a decrease, and a decrease followed by an increase during fruit development and postharvest, respectively. However, the expression levels of these two genes remained relatively stable throughout the latter three periods of Renong-1. The expression levels of Mi20g08560 and Mi05g14240 exhibited a declining trend throughout the development and postharvest in both varieties, and they were higher in Tainong-1 than in Renong-1 at most stages. The expression levels of Mi08g17750 and Mi04g18430 in Renong-1 were consistently higher than those in Tainong-1 across most stages, with a significant decrease observed during the postharvest stage, indicating that Mi08g17750 and Mi04g18430 might be involved in the inhibition of mango fruit maturation.

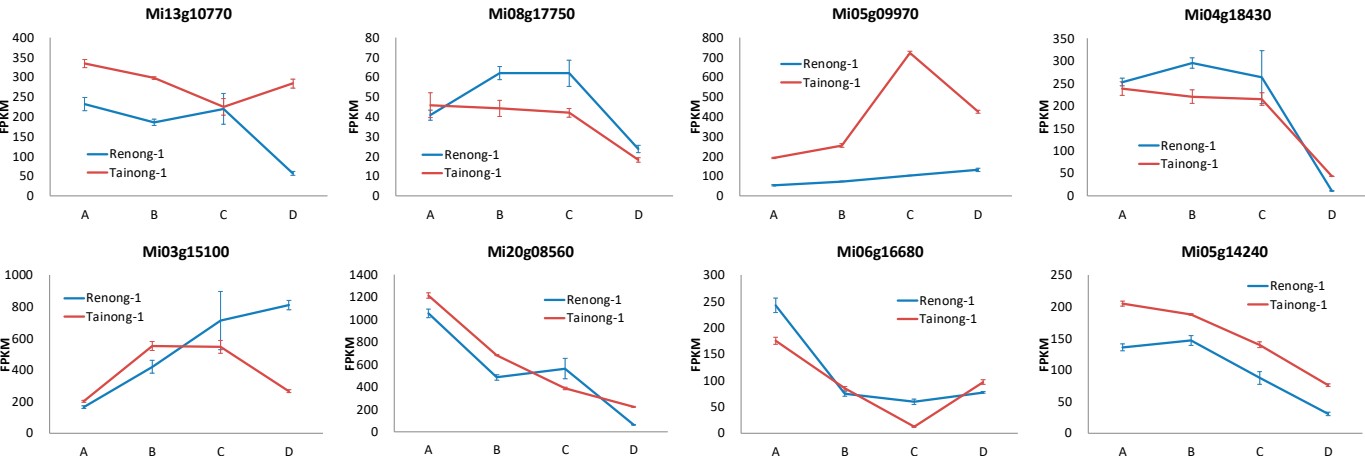

**Figure 6.** Expression of eight key MiMADS genes related to mango ripening in different developmental stages of two varieties. A, B, C, and D represent four different stages (young fruit stage, fruit enlargement stage, green mature stage, and full-ripe stage) of two varieties.

### 3.7. Protein Interaction Network Analysis of the MiMADS Genes

Protein–protein interactions are crucial for the functioning of MADS-box proteins; therefore, evaluating the interaction networks is valuable for characterizing gene function mechanisms. In this study, Mi08g17750 and Mi04g18430 were selected to evaluate possible protein interactions using the STRING website to construct the protein interaction networks. *Arabidopsis thaliana* and *Solanum lycopersicum* were chosen as reference organisms, respectively (Figure 7). When *Arabidopsis thaliana* was used as a reference, both Mi04g18430 and Mi08g17750 were shown to be highly homologous with AGL2, and they may interact with several other MiMADS proteins, including Mi03g15090, Mi20g08570, Mi18g07490, Mi18g11770, Mi19g11260, Mi06g16680, Mi04g18420, Mi05g08820, Mi05g14240, Mi09g07820, Mi18g00050, Mi18g00060, and Mi15g08080. When *Solanum lycopersicum* was taken as a reference, Mi04g18430 and Mi08g17750 were highly homologous with MADS-RIN and TM29, respectively. Both genes may interact with Mi19g11260, Mi18g11770, Mi04g18420, Mi18g00050, Mi18g00060, Mi18g07490, Mi03g15090, Mi20g08570, Mi05g08820, Mi06g16680, Mi05g14240, Mi09g07820, and Mi15g08080. It is worth noting that the screened proteins interacting with Mi04g18430 and Mi08g17750 are consistent when *Arabidopsis thaliana* and *Solanum lycopersicum* were taken as references, respectively.

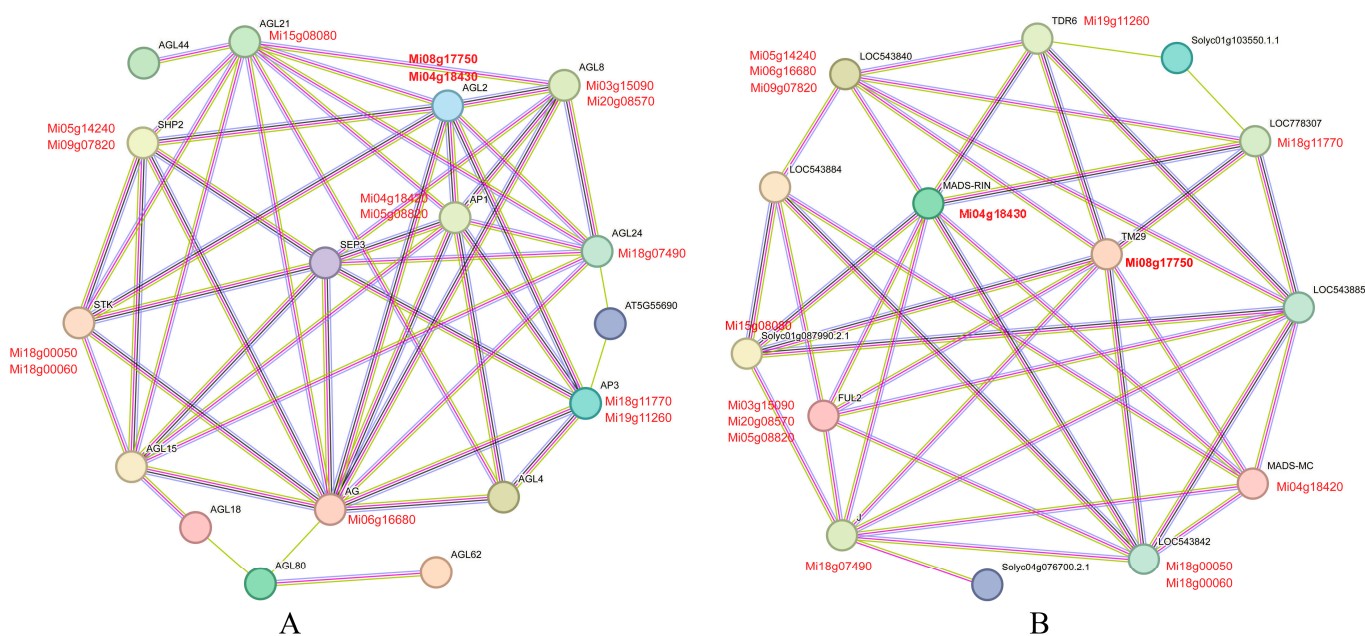

**Figure 7.** Functional interaction network of MiMADS proteins in mango based on their orthologs in *Arabidopsis thaliana* (**A**) and *Solanum lycopersicum* (**B**). Circle nodes in the network represent proteins, and straight lines connecting the nodes represent interaction relationships. Indigo blue and purple lines indicate known interactions; green, red, and navy blue lines indicate predicted interactions; yellow, black, and pearl blue lines indicate texting, co-expression, and protein homology interactions.

## 4. Discussion

The MADS-box genes are ubiquitous in plants, and they play crucial roles in various processes, including plant growth and development, stress response, signal transduction, and other processes. With the publication of high-quality genomes, numerous MADS-domain proteins have been extensively identified and characterized through comprehensive genome-wide identification and analysis of expression profiles, such as Arabidopsis (107 genes) [4], rice (75 genes) [65], tomato (131 genes) [66], potato (153 genes) [67], bread wheat (300 genes) [68], cucumber (43 genes) [69], *Brassica rapa* (167 genes) [70], radish (144 genes) [71], sesame (57 genes) [72], watermelon (39 genes) [73], moso bamboo (42 genes) [74], sorghum (65 genes) [75], apple (146 genes) [76], chrysanthemum (108 genes) [77], eggplant (*Solanum melongena*) (120 genes) [78], blueberry (249 genes) [79], chayote (*Sechium edule*) (70 genes) [80], poplar (105 genes) [81], pear (95 genes) [82], banana (96 genes) [83], grape (58 genes) [84], soybean (106 genes) [85], honeysuckle (*Lonicera japonica*) (48 genes) [86], *Zea mays* (211 genes) [87], and *Rhododendron ovatum* (77 genes) [88].

In this study, a total of 84 MiMADS genes were identified in the mango genome, comprising 22 type I genes and 62 type II genes based on their phylogenetic relationships. Compared with previous studies, it was observed that the number of MADS-box genes varies across different species, but the number of members in type II exceeded that in type I, and this finding was consistent with the results obtained in the present study. Moreover, the number of MADS proteins in different subfamilies varies significantly among species. We found that while mango has a lower overall count of MADS proteins compared to *Arabidopsis thaliana*, there is a higher number of Mα MADS-box genes (16 genes) in mango than in *Arabidopsis thaliana* (11 genes) [4]. This difference may be attributed to gene segmental duplication during the evolutionary process of mango.

In this study, a total of 66 MiMADS proteins were predicted to localize in the nucleus, which is consistent with previous reports that transcription factor proteins mainly participate in regulating specific genes within the eukaryotic nucleus. Additionally, we identified 18 MiMADS proteins residing outside the nucleus, among which eight were found in chloroplasts and three were located in mitochondria. Furthermore, cytoplasmic

localization was predicted for seven proteins. Notably, both chloroplasts and mitochondria contain DNA and RNA, and they also possess essential machinery for gene transcription. It is plausible that certain MiMADS genes may play a regulatory role in photosynthesis and respiratory-related genes. Moreover, some transcription factor proteins exhibit nucleo-cytoplasmic shuttling through phosphorylation/dephosphorylation-mediated trafficking mechanisms [89], suggesting that MiMADS outside the nucleus might translocate into the nucleus when required for gene regulation. Similar findings have been reported regarding MADS protein localization in blueberry [79].

The gene structure, domain, and motif analysis showed that there were differences in the MADS gene of mango. Type II MiMADS genes exhibited a higher number of exons compared to type I genes, indicating their more intricate structural composition. Notably, the exon–intron patterns observed in both type I and type II genes remain conserved across diverse plant taxa, including Arabidopsis [4], rice [67], and grape [90], underscoring the high conservation of MADS-box TFs among plants. However, it is worth mentioning that several type I group genes also possess two or more intron structures, indicating these genes might have different functions. Different subfamilies exhibit variations in the number and types of domains and conserved motifs [91]. In our study, it was observed that some MIKC$^C$ genes and all MIKC* genes lack K-box domains, whereas certain type I MiMADS genes contain K-box domains. The absence of K-box domains in the MIKC* subfamily genes has been reported in several species, including foxtail millet [92], American beautyberry [93], and litchi [94].

Some studies have indicated that intron loss and insertion mutations are commonly observed during the evolutionary process of plant MADS-box genes [71]. MIKC$^C$ genes are believed to be the most ancient members of the MADS-box gene family, with type I genes likely evolving from MIKC$^C$ genes [5]. This suggests that MIKC* genes may be a class of transition genes retained after the loss of the K-box domain during the evolution of MIKC$^C$ genes. Furthermore, certain type I MiMADS genes retain conserved K-box domains throughout their evolution. Additionally, it is worth noting that the MiMADS protein motifs within the same subfamily exhibit variations, indicating that intron loss or gain may serve as a pattern for MiMADS gene evolution and contribute significantly to the functional diversity within the MiMADS family.

The MADS-box genes are believed to originate from gene duplication events that initially occurred in the most recent common ancestor of extant eukaryote lineages [95]. Gene replication plays an important role in the amplification and evolution of transcription factors. In this study, 34 MiMADS genes (40.48%) were located in 14 clusters as tandem duplicated genes, and 47 pairs of MiMADS genes were found on segmental duplicated genome blocks. These findings highlight the significant contributions of both tandem duplication and segmental replication in facilitating the expansion of MADS-box genes within mango genomes. The same results were observed in *Cyclocarya paliurus* [96], *Rhododendron hainanense* Merr. [97], *Fagopyrum tataricum* [98], Arabidopsis [4], and potato [71]. However, only tandem duplications have been documented in cucumber [73]. These observations suggest that gene duplication plays diverse roles across different species. In addition, we observed that with the exception of seven pairs of segmental replication gene pairs with an undefined k value (NA), all other forty pairs exhibited a k value less than 1, indicating a significant role of purifying selection in the segmental duplication events within the MiMADS gene family. These findings align with previous studies, indicating that protein neofunctionalization may not be the predominant status for the retained genes from recent WGD of mango [52].

Through collinearity analysis among different species, a total of 60 pairs of collinear MADS-box genes were identified between mango and Arabidopsis, while 89 pairs were found between mango and *Citrus sinensis* (sweet orange). The number of homologous events observed between mango and *Citrus sinensis* was significantly higher compared to that between mango and Arabidopsis, which is consistent with the relatively closer evolutionary relationship between mango and *Citrus sinensis* (sweet orange).

We identified a total of 287 cis-acting elements associated with phytohormones, which were found to be present in 76 MiMADS genes. Furthermore, we observed the presence of cis-acting elements related to stress responses, including anaerobic conditions, low temperatures, and wound signals in 77 MiMADS genes. These findings strongly suggest that the majority of MiMADS genes are under hormonal regulation and play crucial roles in the growth, development, and stress resistance mechanisms of mango fruit.

According to the RNA-seq data and phylogenetic tree analysis, we hypothesized that Mi20g08560, Mi03g15100, Mi05g09970, Mi13g10770, Mi04g18430, and Mi08g17750 of the SEP1 subfamily along with Mi05g14240 and Mi06g16680 of the AG subfamily were key MADS genes associated with mango ripening. Moreover, both Mi08g17750 and Mi04g18430 played a role in inhibiting mango fruit maturation, which aligns with previous findings [43]. The expression of four SEP genes in bananas has been shown to elevate ethylene content and influence banana maturation [99]. Similarly, the expression of SEP4-like genes is essential for strawberry ripening [100].

However, due to limitations in the selection of genomic data, experimental materials, and analysis software methods, all of these results are only predictive based on data analysis, and they need to be further validated by other experiments in the future.

**5. Conclusions**

In this study, 84 MADS-box gene family members were identified in the mango genome, and their physical and chemical properties, subcellular localization, chromosome position, gene structure, protein conserved domains, motifs, evolutionary connections, and cis-acting elements were analyzed. The expression profiles of MADS genes in different fruit development stages of two varieties were also analyzed. Our findings lay the foundation for a comprehensive functional characterization of the MADS-box gene family in mango and provide candidate genes for studying the role of the MADS-box gene family in regulating fruit ripening. Additionally, the potential functions and regulatory mechanisms of Mi08g17750 and Mi04g18430 will be experimentally illuminated in our future research. This study will help us to comprehensively understand the characteristics of the MiMADS gene family and screen for MiMADS genes involved in mango fruit ripening.

**Supplementary Materials:** The following supporting information can be downloaded at: https://www.mdpi.com/article/10.3390/horticulturae9121289/s1, Figure S1. Gene structure analysis of MiMADS genes; Figure S2. Synteny analysis of MADS-box genes between *Mangifera indica*, *Citrus sinensis*, and *Arabidopsis thaliana*. The blue lines indicate MADS collinearity; Figure S3. Phylogenetic tree analysis of the screened differentially expressed MADS proteins and other reported maturity-related MADS proteins; Table S1: Protein information analysis of MADS-box gene family members in mango; Table S2: The Ka and Ks values for 47 duplicated gene pairs of MADS-box gene family members in mango; Table S3: Details of the cis-acting elements for MADS-box gene family members in mango.

**Author Contributions:** Methodology, B.Z. and H.W.; project administration, S.W., X.M. and J.F.; resources, S.W. and H.W.; supervision, S.W., H.W. and J.F.; validation, B.Z.; visualization, W.X. and K.X.; writing—original draft, B.Z.; writing—review and editing, L.S. and J.F. All authors have read and agreed to the published version of the manuscript.

**Funding:** This work was supported by the Natural Science Foundation of Hainan Province (322QN378), Fund for Less Developed Regions of the National Natural Science Foundation of China (32360727), Key Research and Development Program of Hainan Province (ZDYF2022XDNY255), Seed Industry Revitalization Project of Guangdong Province (2022-NPY-00-030).

**Data Availability Statement:** MADS-box gene members of *Arabidopsis thaliana* were downloaded from the TAIR database (http://www.arabidopsis.org/ accessed on 22 April 2022). MADS-box gene members of *Citrus sinensis* were downloaded from citrus (http://citrus.hzau.edu.cn/orange/ accessed on 10 May 2022) databases. The transcriptome datasets of mango were deposited in the Genome Sequence Archive (SRA) database of NCBI under the accession number PRJNA 629065 (accessed on 22 May 2022).

**Acknowledgments:** We thank the anonymous reviewers for their helpful comments and suggestions, which allowed us to improve the quality of this manuscript.

**Conflicts of Interest:** The authors declare no conflict of interest.

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
