# Peer review of "Genome-Wide Identification of Mango (Mangifera indica L.) MADS-Box Genes Related to Fruit Ripening"

_horticulturae, doi:10.3390/horticulturae9121289_

Round 1

Reviewer 1 Report (New Reviewer)

Comments and Suggestions for Authors

The article provides a significant contribution to understanding the molecular mechanisms in fruit maturation, particularly in mangoes, a fruit of considerable economic value. The discovery of 84 MADS-box genes, including 22 type I and 62 type II, not only enriches genetic knowledge but also highlights the gene family’s complexity in this tropical fruit. The meticulous gene duplication analysis clarifies how tandem duplication and segmental replication have contributed to the MADS-box genes' expansion in the mango genome, indicating the importance of purifying selection in maintaining gene function through evolution. Furthermore, the investigation into cis-regulatory elements offers insights into hormonal regulation and stress resistance in mango fruit development. Identifying six SEP1 subfamily MiMADS genes and two AG subfamily genes as key regulators of mango maturation pinpoints potential targets for genetic engineering or marker-assisted breeding in mango improvement programs. The study of Mi08g17750 and Mi04g18430 genes from the SEP1 subfamily as primary regulators in fruit maturation inhibition, and their interaction networks, shows direct research application to practical uses. These findings are valuable for advancing fruit biotechnology and developing maturation manipulation strategies, vital for the food and agriculture industry. In conclusion, this exemplary article merges sophisticated genetic analyses with practical application, not only filling a significant knowledge gap but also laying a strong foundation for future regulatory mechanism investigations in mango maturation. I recommend this article for publication in a high-impact scientific journal due to its scientific relevance and potential applicability in fruit improvement.

Below I put some questions for you to present simple changes, arguments or discussions about:

Identification and Analysis of MADS-Box Genes in Mango (Section 2.1)

Genomic Data Sources: The study relies on a single source for the mango genomic sequence (Wang [50]), which could limit the analysis if there are sequencing or annotation errors in this specific source. Diversifying the sources of genomic data could increase the robustness of the results.

Trans-specific Comparison: Using Arabidopsis MADS-box genes as a reference to identify homologs in mango is common, but may not capture all the particularities of mango-specific MADS-box genes. The divergent evolution and unique functionality of some of these genes may not be identified by this approach alone.

HMMER and BLAST Search Parameters: The e-value threshold parameters can influence the number of genes identified. An e-value of 1e-5 is relatively conservative, but could also exclude truly positive genes that have a weaker similarity to the known domains.

Domain Analysis: Manual verification of genes without the MADS domain may introduce bias if curation is not rigorously standardized.

Phylogenetic Tree, Gene Structure, and Analysis of Conserved Motifs (Section 2.2)

Phylogenetic Construction: The maximum likelihood method is robust, but the results may be sensitive to the chosen evolutionary models and the sequence alignment algorithms used.

Gene Structure and Motif Analysis: The identification of gene structures and conserved motifs may be limited by the quality of sequence alignment and the accuracy of the prediction tools used.

Chromosomal Distribution, Gene Duplication, and Ka/Ks Analysis (Section 2.3)

Linkage Maps and Chromosomal Localization: The accuracy of chromosomal localization of genes depends on the precision of the genetic map and the completeness of the reference genome.

Gene Duplication Analysis: The inference of duplication events may not reflect complex scenarios of genome evolution, such as aneuploidy, translocations, or gene conversions.

Analysis of Cis Elements in MiMADS (Section 2.4)

Prediction of Cis-regulatory Elements: This analysis is dependent on the accuracy of the PlantCARE database and may not detect new cis-regulatory elements that have not yet been characterized or are unique to mango.

Interaction Network Analysis (Section 2.5)

Protein Interaction Analysis: Using Arabidopsis and tomato models may not be entirely representative of protein interactions in mango, due to differences in species biology.

Plant Material Collection and Expression Analyses (Section 2.6)

Genetic Variation: Different mango varieties may present genetic variations that influence the expression of MADS-box genes, and the article does not discuss how this variation might affect the results.

RNA-seq Data Quality Control: Although the article mentions the removal of low-quality reads, it is not clear how the reads were validated and quantified, which could influence the identification of differentially expressed genes.

Author Response

Responses to Reviewer's comments:

Reviewer #1:

Comments: The article provides a significant contribution to understanding the molecular mechanisms in fruit maturation, particularly in mangoes, a fruit of considerable economic value. The discovery of 84 MADS-box genes, including 22 type I and 62 type II, not only enriches genetic knowledge but also highlights the gene family’s complexity in this tropical fruit. The meticulous gene duplication analysis clarifies how tandem duplication and segmental replication have contributed to the MADS-box genes' expansion in the mango genome, indicating the importance of purifying selection in maintaining gene function through evolution. Furthermore, the investigation into cis-regulatory elements offers insights into hormonal regulation and stress resistance in mango fruit development. Identifying six SEP1 subfamily MiMADS genes and two AG subfamily genes as key regulators of mango maturation pinpoints potential targets for genetic engineering or marker-assisted breeding in mango improvement programs. The study of Mi08g17750 and Mi04g18430 genes from the SEP1 subfamily as primary regulators in fruit maturation inhibition, and their interaction networks, shows direct research application to practical uses. These findings are valuable for advancing fruit biotechnology and developing maturation manipulation strategies, vital for the food and agriculture industry. In conclusion, this exemplary article merges sophisticated genetic analyses with practical application, not only filling a significant knowledge gap but also laying a strong foundation for future regulatory mechanism investigations in mango maturation. I recommend this article for publication in a high-impact scientific journal due to its scientific relevance and potential applicability in fruit improvement.

Response: Thank you for your affirmation of our manuscript. We really appreciate your sincere comments to this manuscript. We have carefully revised the manuscript according to your suggestions.

Specific comments:

Identification and Analysis of MADS-Box Genes in Mango (Section 2.1)

  • Genomic Data Sources: The study relies on a single source for the mango genomic sequence (Wang [50]), which could limit the analysis if there are sequencing or annotation errors in this specific source. Diversifying the sources of genomic data could increase the robustness of the results.

Response: Thank you for the suggestions. Though diversifying the sources of genomic data could increase the robustness of the results, the published genomic data are derived from different mango varieties, and the differences in MADS-box proteins between mango varieties will greatly interfere with subsequent analysis of MADS-Box gene family. So we chose a genomic data that is currently recognized as having the best assembled quality in our study. We also added a discussion of the limitations of this study in the discussion of our revised manuscript.

  • Trans-specific Comparison: Using Arabidopsis MADS-box genes as a reference to identify homologs in mango is common, but may not capture all the particularities of mango-specific MADS-box genes. The divergent evolution and unique functionality of some of these genes may not be identified by this approach alone.

Response: Thank you for the suggestions. In order to maximize the identification of MADS family proteins in genomic data, HMMER Search was also performed.

  • HMMER and BLAST Search Parameters: The e-value threshold parameters can influence the number of genes identified. An e-value of 1e-5 is relatively conservative, but could also exclude truly positive genes that have a weaker similarity to the known domains.

Response: Thank you for the suggestions. We also carefully considered the threshold parameters for a long time, and after consulting several references, we ultimately decided to set it at 1e-5. We will adjust the e-value threshold parameters in future studies according to your suggestions.

  • Domain Analysis: Manual verification of genes without the MADS domain may introduce bias if curation is not rigorously standardized.

Response: Thank you for the suggestions. We apologize for the ambiguity caused by our description of this sentence in our manuscript, we have revised this sentence in our revised manuscript.

Phylogenetic Tree, Gene Structure, and Analysis of Conserved Motifs (Section 2.2)

  • Phylogenetic Construction: The maximum likelihood method is robust, but the results may be sensitive to the chosen evolutionary models and the sequence alignment algorithms used.

Response: Thank you for the suggestions. The analysis was also performed using Neighbor-Joining method. Finally, according to the results of the two methods, we choose to use the maximum likelihood method.

  • Gene Structure and Motif Analysis: The identification of gene structures and conserved motifs may be limited by the quality of sequence alignment and the accuracy of the prediction tools used.

Response: Thank you for the suggestions. We have added a discussion of the limitations of this study in the discussion of our revised manuscript.

Chromosomal Distribution, Gene Duplication, and Ka/Ks Analysis (Section 2.3)

  • Linkage Maps and Chromosomal Localization: The accuracy of chromosomal localization of genes depends on the precision of the genetic map and the completeness of the reference genome.

Response: Thank you for the suggestions. We have added a discussion of the limitations of this study in the discussion of our revised manuscript.

  • Gene Duplication Analysis: The inference of duplication events may not reflect complex scenarios of genome evolution, such as aneuploidy, translocations, or gene conversions.

Response: Thank you for the suggestions. we have revised this in our revised manuscript.

Analysis of Cis Elements in MiMADS (Section 2.4)

  • Prediction of Cis-regulatory Elements: This analysis is dependent on the accuracy of the PlantCARE database and may not detect new cis-regulatory elements that have not yet been characterized or are unique to mango.

Response: Thank you for the suggestions. PlantCARE database is a recognized database for predicting cis-regulatory elements. This study only carried out preliminary prediction analysis of cis-regulatory elements, and we will analyze specific cis-regulatory elements of mango in the future.

Interaction Network Analysis (Section 2.5)

  • Protein Interaction Analysis: Using Arabidopsis and tomato models may not be entirely representative of protein interactions in mango, due to differences in species biology.

Response: Thank you for the suggestions. The protein interaction analysis in this study is just preliminary prediction analysis. In order to improve the accuracy of the analysis, both Arabidopsis and tomato models were chosen. The protein interactions in mango needs to be further verified by other experiments.

Plant Material Collection and Expression Analyses (Section 2.6)

  • Genetic Variation: Different mango varieties may present genetic variations that influence the expression of MADS-box genes, and the article does not discuss how this variation might affect the results.

Response: Thank you for the suggestions. We have added a discussion of the limitations of this study in the discussion of our revised manuscript.

  • RNA-seq Data Quality Control: Although the article mentions the removal of low-quality reads, it is not clear how the reads were validated and quantified, which could influence the identification of differentially expressed genes.

Response: Thank you for the suggestions. In this study, we selected the transcriptome raw data that we had previously published, which has been validated using QRT-PCR. Moreover, based on this RNA-seq Data, we obtained very good results in this study, which consistent with some previous reports.

Reviewer 2 Report (New Reviewer)

Comments and Suggestions for Authors

This is a good research, well undertaken and well written and is worthy of publication as it will add to our understanding of MADS box genes and their regulation in a perennial crop spp. My major comment is the excessive use of figures in the manuscript. In my view some of these figures which are not central to the message delivered can be put in as supplementary file s only those who want it will access it. I would strongly suggest moving Figs 3, 5 and 8 to Supplementary data, as that information is not central and also dilutes the message and distracts the read flow.

Introduction can be shortened and more focused as there is quite a bit if general description that goes with it. I will leave it to the Editor and the authors as this is not as important as the results.

Figure Legends are way too brief. A bit more descriptive legends will be good to get the message from the figure directly instead of reading the relevant results from the result section. This is quite important.

Other minor editorial fixes:

Line 34: change to MADS box TFs from MADS TFs.

36: Expand MIKC- I didn't see it expanded anywhere. There are a few other abbreviations like that in the ms that needs to be addressed as well

74-75: Mango is a significant fruit crop of the world, not just south China. Change it to reflect it as a global fruit crop

121 and down: There are several words/phrases in RED and not sure if this signifies anything. Take care of those and make them all black.

211-212. Delete 'with most seven to eight exon containing genes' as it is too convoluted and redundant.

218: replace duplication with changes to avoid repetition

365: 3.7 and Fig 10. I think the figure 10 should be part of these results but is not mentioned there. Please check.

References are a bit excessive and can be pruned down a bit. If the Introduction is more focused then that will take care of it. I leave that to he Editor

Comments on the Quality of English Language

Generally fine and I didn't see any major issues with the language

Author Response

Responses to Reviewer's comments:

Comments: This is a good research, well undertaken and well written and is worthy of publication as it will add to our understanding of MADS box genes and their regulation in a perennial crop spp.

Response: Thank you for your affirmation of our manuscript. We really appreciate your sincere comments to this manuscript. We have carefully revised the manuscript according to your suggestions.

Specific comments:

  • My major comment is the excessive use of figures in the manuscript. In my view some of these figures which are not central to the message delivered can be put in as supplementary file s only those who want it will access it. I would strongly suggest moving Figs 3, 5 and 8 to Supplementary data, as that information is not central and also dilutes the message and distracts the read flow.

Response: Thank you for the suggestions. We have moved these figures to Supplementary data in our revised manuscript.

  • Introduction can be shortened and more focused as there is quite a bit if general description that goes with it. I will leave it to the Editor and the authors as this is not as important as the results.

Response: Thank you for the suggestions. We have shortened the introduction in our revised manuscript.

  • Figure Legends are way too brief. A bit more descriptive legends will be good to get the message from the figure directly instead of reading the relevant results from the result section. This is quite important.

Response: Thank you for the suggestions. We have detailed the figure legends of some figures in our revised manuscript.

Other minor editorial fixes:

  • Line 34: change to MADS box TFs from MADS TFs.

Response: Thank you for the suggestions.

  • 36: Expand MIKC- I didn't see it expanded anywhere. There are a few other abbreviations like that in the ms that needs to be addressed as well

Response: Thank you for the suggestions. We have expanded MIKC and other abbreviations in our revised manuscript.

  • 74-75: Mango is a significant fruit crop of the world, not just south China. Change it to reflect it as a global fruit crop

Response: Thank you for the suggestions. We have revised this sentence in our revised manuscript.

  • 121 and down: There are several words/phrases in RED and not sure if this signifies anything. Take care of those and make them all black.

Response: Thank you for the suggestions. We have revised the color of these words in our revised manuscript.

  • 211-212. Delete 'with most seven to eight exon containing genes' as it is too convoluted and redundant.

Response: Thank you for the suggestions. We have revised this sentence in our revised manuscript.

  • 218: replace duplication with changes to avoid repetition

Response: Thank you for the suggestions. The word “duplication” a special word, the word “changes” does not convey the exact meaning of the word “duplication”. We haven’t revised this sentence in our revised manuscript.

  • 365: 3.7 and Fig 10. I think the figure 10 should be part of these results but is not mentioned there. Please check.

Response: Thank you for the suggestions. We have marked it in our revised manuscript.

  • References are a bit excessive and can be pruned down a bit. If the Introduction is more focused then that will take care of it. I leave that to the Editor.

Response: Thank you for the suggestions. Each of the listed references has certain significance for the manuscript. I am very sorry for not deleting the references. Thanks again for your suggestions.

Comments on the Quality of English Language

  • Generally fine and I didn't see any major issues with the language

Response: Thank you for the affirmation of English language.

This manuscript is a resubmission of an earlier submission. The following is a list of the peer review reports and author responses from that submission.

Round 1

Reviewer 1 Report

Comments and Suggestions for Authors

Reviewer comments:

The manuscript entitled “Genome-Wide Identification of Mango (Mangifera indica L.) MADS-Box Genes Related to Fruit Ripening.” by Zheng et al. I found this topic interesting, demonstrates an MADS-box genes involved in mango fruit ripening. But I have few concerns related to the research article. I am asking authors to revise the manuscript carefully considering my comments for possible publication in “Horticulturae”.

I have given my comments.

• The present investigation will be a good contribution to the genetic improvement of Mango.

• Line No 121: The authors requested to check and correct “Gene Duplication, and Ka/Ks Analysis of MYBs”.

• Line No 178: The authors requested to check and correct “MiMYB proteins”.

• Line No 383 to 386: Authors should check and remove the sentence “Authors should discuss the results and how they can be interpreted from the perspective of previous studies and of the working hypotheses. The findings and their implications should be discussed in the broadest context possible. Future research directions may also be highlighted”.

• Line No 326: Authors must elaborate the figure legends for provided figures.

The submitted manuscript may be acceptable for publication after a major revision.

Author Response

Comments:

The manuscript entitled “Genome-Wide Identification of Mango (Mangifera indica L.) MADS-Box Genes Related to Fruit Ripening.” by Zheng et al. I found this topic interesting, demonstrates an MADS-box genes involved in mango fruit ripening. But I have few concerns related to the research article. I am asking authors to revise the manuscript carefully considering my comments for possible publication in “Horticulturae”. The submitted manuscript may be acceptable for publication after a major revision.

Response: Thank you for giving us the opportunity to revise the manuscript. We sincerely appreciate your comments on this manuscript. We have carefully made revisions to the manuscript based on your suggestions.

Specific comments:

  • The present investigation will be a good contribution to the genetic improvement of Mango.

Response: We sincerely appreciate your recognition of our research. Thank you for your comments.

  • Line No 121: The authors requested to check and correct “Gene Duplication, and Ka/Ks Analysis of MYBs”.

Response: Thank you for the comments. We are sorry for this clerical error in the manuscript and have carefully revised the full manuscript. Thanks again for your suggestions.

  • Line No 178: The authors requested to check and correct “MiMYB proteins”.

Response: Thank you for the comments. We apologize for the clerical error in the manuscript and have thoroughly revised it. Thank you once again for your valuable suggestions.

  • Line No 383 to 386: Authors should check and remove the sentence “Authors should discuss the results and how they can be interpreted from the perspective of previous studies and of the working hypotheses. The findings and their implications should be discussed in the broadest context possible. Future research directions may also be highlighted”.

Response: Thank you for the comments. We are really sorry for this typographical error in this manuscript due to our carelessness. We've already removed that sentence. Thanks again for your suggestions.

  • Line No 326: Authors must elaborate the figure legends for provided figures.

Response: Thank you for your suggestion. We have added a note to provided figures. Please refer to line No 326-328, No 363-365, and No 384-388. Thank you again for your suggestions.